# FAILURE MODES OF LLMS FOR CAUSAL REASONING ON NARRATIVES

## ABSTRACT

In this work, we investigate the causal reasoning abilities of large language models (LLMs) through the representative problem of inferring causal relationships from narratives. We find that even state of the art language models rely heavily on unreliable shortcuts, both in terms of the narrative presentation and their parametric knowledge. For example, LLMs tend to determine causal relationships based on the topological ordering of events (i.e., earlier events cause later ones), resulting in lower performance whenever events are not narrated in their exact causal order. Similarly, we demonstrate that LLMs struggle with long-term causal reasoning — they often fail when the narratives are longer and contain many events. As an additional failure mode, we show LLMs appear to heavily rely on their parametric knowledge at the expense of reasoning over the provided narrative. This degrades their abilities whenever the narrative opposes parametric knowledge. We extensively validate these failure modes through carefully controlled synthetic experiments, as well as evaluations on real-world narratives. Finally, we observe that explicitly generating a causal graph generally improves performance while naive chain-of-thought is ineffective. Collectively, our results distill precise failure modes of current state-of-the art models and can pave the way for future techniques to enhance causal reasoning in LLMs.

## 1 INTRODUCTION

Causal reasoning is a core component of intelligence and decision-making, enabling an agent to move beyond associations (i.e., events that are observed to occur together) to determine the consequences of their actions (i.e., interventions) and answering *what-if* questions (counterfactuals). Causal models have been widely studied in machine learning and artificial intelligence (Peters et al., 2017; Pearl, 2009) as well as in the context of human cognition (Waldmann, 2017). The advent of Large Language Models (LLMs) has led to opportunities for leveraging large-scale textual data for improving causal inference (Liu et al., 2024; Zhang et al., 2023).

While many works have shown that LLMs are capable of memorizing and recalling causal knowledge, they can fail at reliably leveraging that knowledge for reasoning (Jin et al., 2023; Zečević et al., 2023). For example, when reading a news article or a story, understanding *why* a particular event occurred requires reasoning over the causal structure of the events implied by the given narrative. Similarly, determining which events in the narrative will be affected by certain actions requires reasoning about which events are causally downstream of that action. In such cases, simply recalling parametric causal knowledge (that some event typically causes another event) may not suffice as the agent must leverage the causal structure inherent to the specific narrative at hand.

In this work, we aim to understand the causal reasoning abilities of LLMs from textual narratives. We consider settings where there is an (unknown) underlying causal chain graph of the form $V_1 \to V_2 \to \ldots \to V_N$, where each node $V_i$ has some semantic meaning (e.g., smoking), that is verbalized in the form of a (realistic) narrative $S$. For a given narrative $S$, we consider two kinds of causal reasoning tasks: (1) Does $V_i$ have a causal effect on $V_j$? and (2) Given the node identities $(V_1, \ldots V_N)$, construct a causal chain graph faithful to the narrative. While these tasks do not encompass all aspects of causal reasoning (e.g., counterfactual reasoning), they are important primitives for successful causal reasoning.

Our primary contribution is to shed light on the limitations and unreliable shortcuts that LLMs use for these two causal reasoning tasks. We focus on the following failure modes. Firstly, we show that LLMs heavily rely on the order in which the causal relationships are verbalized in the narrative. We observe that when the narrative is constructed in the reverse topological order of the causal chain (i.e., edge $V_i \rightarrow V_{i+1}$ is narrated before $V_{i-1} \rightarrow V_i$), the performance of the LLM suffers as it often assigns the cause to an earlier event and the effect to a later event in the narrative. Secondly, we show that LLMs use their parametric causal knowledge (i.e., if an event typically causes another event) as a shortcut to answer causal questions. Thus, when the cause-and-effect pairs implied by the narrative conflict with the parametric knowledge, the LLM often ignores the specifics of the narrative and defaults to its parametric knowledge. This occurs even when the narratives are written to explicitly disagree with parametric knowledge and the LLM is prompted to ignore outside information. Thirdly, we find that LLMs fail more often as the narratives become longer.

Perhaps surprisingly, we also find that the LLM's answers to cause-and-effect queries, including with Chain of Thought (CoT) (Wei et al., 2022) and In-Context Learning, do not have a high degree of consistency with the causal graph that the LLM extracts from the narrative. On the other hand, we observe that just using the extracted graph to make decisions often seems to alleviate the above failure modes. This finding is notable as it suggests that the specific formatting of the causal graph extraction task can more reliably elicit the language model's causal reasoning capabilities relative to the baseline of CoT prompting.

We first validate our findings through a series of carefully controlled experiments with synthetic causal events and narratives (Sec. 3). We use the LLM to generate a causal chain of events that form the ground truth causal graph. We then prompt the LLM to generate a narrative-based on that causal graph. Next, we conduct experiments on narratives generated from real-world causal chain graphs (Sec. 4) extracted from *CauseNet* (Heindorf et al., 2020), a dataset of (claimed) causal relationships between real-world concepts. We generate narratives for these chain graphs using the LLM as well as from sentences in *CauseNet*, ensuring that the narrative as a whole remains coherent.

Leveraging the LLM to verbalize causal relationships leads to more natural-sounding narratives (which is difficult with formulaic templates). Depending on the specific causal relationship, causality might be indicated using a phrase like '*causes*', '*leads to*', '*resulted in*', etc. We focus on simple chain graphs to isolate the simplest cases of the failure modes, demonstrating that the LLMs take unreliable shortcuts on these simple graphs, both when the narratives are generated by the LLM itself as well as narratives with real-world sentences.

Taken together, our work makes the following contributions:

1. We demonstrate that even *state-of-the-art* large language models are unreliable at extracting causal relationships from realistic narratives expressing simple chain graphs.

2. Focusing on two key aspects of causal reasoning, we distill concrete failure modes to account for this unreliability: reliance on positional shortcuts, parametric knowledge, and narrative length.

3. We investigate the impact of various prompting strategies on reasoning ability, finding that *incorporating the estimated causal graph structure* is most helpful.

## 2 RELATED WORKS

**Reasoning in Large Language Models**   Recently, there has been significant interest in evaluating and improving the reasoning capabilities of large language models. Prior works have examined reasoning in diverse settings including mathematics (Cobbe et al., 2021; Hendrycks et al., 2021), social interactions (Sap et al., 2019),and common-sense tasks (Zellers et al., 2019). Through this extensive investigation, large language models have demonstrated unreliable reasoning capabilities, often failing unexpectedly on relatively straightforward queries (Wan et al., 2024; Nezhurina et al., 2024). Extensive prior work has also studied methods for improving the reasoning capabilities of large language models such as by eliciting step-by-step explanations (Wei et al., 2023), finetuning on reasoning traces (Zelikman et al., 2022), and training models to find and correct errors (Kumar et al., 2024). In this work, we perform a comprehensive study of the causal reasoning capabilities of large-language models, identifying the existence of key failure modes in state-of-the-art models.

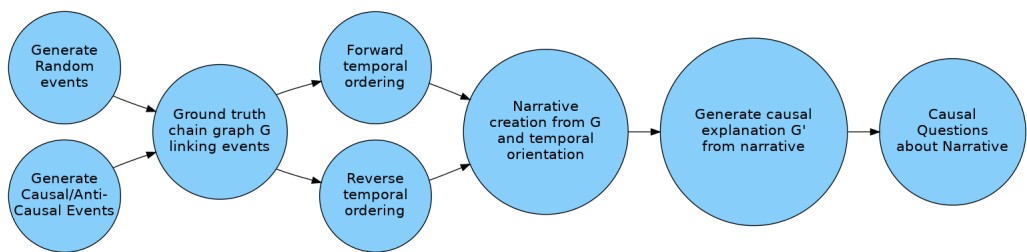

Figure 1: Summary of Synthetic Experiment Setup

**Causal Reasoning in Large Language Models** Prior works have also examined the causal capabilities of large-language models. Jin et al. (2023) develops a benchmark for testing the causal reasoning in LLMs given causal graphs, finding that language models can struggle with the task. However, the settings examined in Jin et al. (2023) require computations based on probabilities which is known to be challenging for LLMs. Zečević et al. (2023) hypothesizes that large language model may only be "imitating" causal reasoning abilities from their pretraining corpus. Their work additionally investigates what they call "natural word chains," or how well the LLM is able to accurately identify causation of events when they are linked together in chain graphs. Our paper builds on this by performing various tests on narratives that correspond to graphs. Kıcıman et al. (2024) shows that LLMs have strong abilities to generate causal texts. We make use of this generation for narrative creation in our synthetic experiments. Tan et al. (2022) shows the capability of a neural network trained on news data to label causal structures in individual sentences.

**Parametric Knowledge Conflicts** Large language models have been shown to memorize factual knowledge present in their pretraining corpora, resulting in *parametric knowledge* (Petroni et al., 2019; Jiang et al., 2020). Prior works, however, have observed that language models can often fail to use information provided in their context as a result of such memorized knowledge (Krishna et al., 2021; Longpre et al., 2022). This has been identified as a significant hurdle in the reliability of summarization and retrieval augmented-based systems (Rehman et al., 2023; Jin et al., 2024). Xie et al. (2024) finds that the coherence of provided context can control the extent to which parametric knowledge is over-utilized. In this work, we examine the distinct setting of causal reasoning and demonstrate that it can also be harmed by over-reliance on parametric knowledge.

## 3 EXPERIMENTS WITH SYNTHETIC DATA

### 3.1 DATA GENERATION PROCESS

In our synthetic experiments, we use two leading LLMs: OpenAI's GPT-4 (OpenAI et al., 2024) and Anthropic's Claude 3.5 Sonnet (Anthropic, 2024). The purpose of our synthetic setup is to carefully control the conditions under which the LLMs are tested. In terms of the general setup of our fully synthetic experiments, as we summarize in Figure 1, we first use the LLM to generate events (which are real world phenomena like *rain* or *plants growing*). Then these events are linked together into a chain graph $G$ that acts as the causal ground truth. The LLM is given $G$ and asked to create a narrative that stays faithful to the causal relationships in $G$.

Providing only the narrative as input (and not $G$), we then ask the LLM to find $G'$, the prediction of the underlying causal structure expressed by the narrative. Next, a series of causal questions is created by randomly sampling 10 tuples of events from $G$ and asking the LLM whether an event in the tuple causes the other based on the narrative and/or $G'$. We also explore many mediums of prompting including Chain of Thought (COT) ((Wei et al., 2023)) and In-Context Learning ((Dong et al., 2024)) (prompts are in Appendix A and the supplement).

### 3.2 RELIANCE ON TOPOLOGICAL ORDERING

Our experiments show that LLMs rely on the ordering in which the events are verbalized in a narrative when determining causal relationships. To investigate this, we started with randomly generated

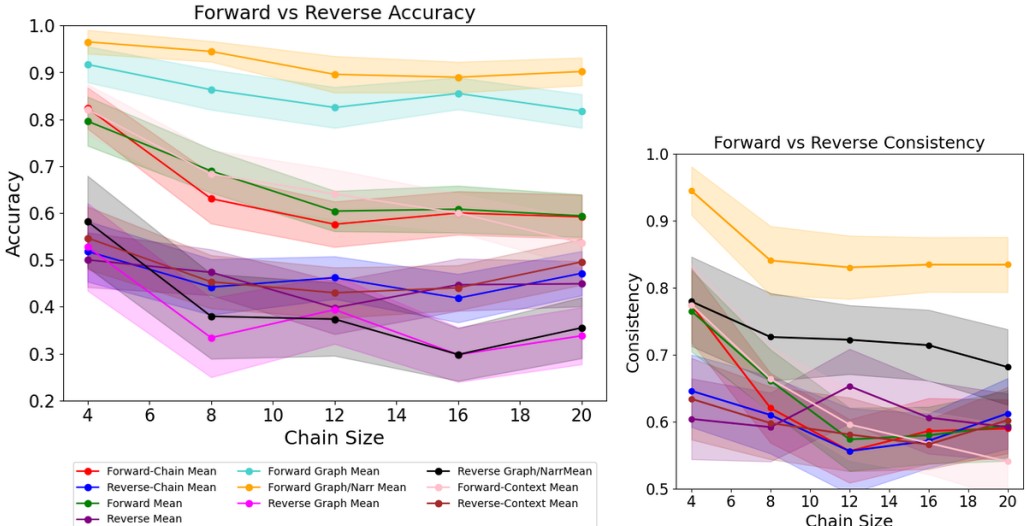

Figure 2: GPT-4 Test of the LLM's ability to reason on narratives written in the Forward and Reverse topological orientations. Chain size is the number of nodes in $G$. Forward/Reverse Chain - COT, Forward/Reverse Mean -simple prompting strategy, Forward/Reverse Graph Mean - using only $G'$, Forward/Reverse Graph/Narr Mean - prompting the LLM to consider both the narrative and $G'$, Forward/Reverse Context Mean - In-Context Learning. Accuracy measures LLM answer agreement with $G$, consistency measures agreement with $G'$. Shaded zone around lines is 95 % CI

events that were used to make a ground truth graph $G$. During the creation of the narrative, we specified that the LLM either place the events in (1) the order that matches the topological causal ordering of the graph (e.g., if event $A$ causes $B$, then event $A$ is mentioned before $B$ in the narrative), or (2) a way that runs opposite to the causal ordering (event $B$ would be mentioned before event $A$ in the narrative even though $A$ causes $B$). We refer to these as the *Forward* and a *Reverse* topological ordering, respectively. Upon inspection of the generated narratives, human evaluators (paper authors who were blinded to the causal graph) were able to properly extract the ground truth causal graph (94% of the time out of 50 randomly sampled narratives). As an example, the following is a GPT-4 generated *Reverse* topological narrative for the causal chain *Film festival → Food truck rally → Trampoline park party*:

> In a *trampoline park party*, guests bounced and laughed under the neon lights. The high-flying event was made possible by the success of the *food truck rally* that took place earlier in the day. Food trucks lined the streets, offering a variety of delicious treats that fueled the fun and energy of the park party attendees. *The film festival* set the stage for the rest of the events, creating a desire for cultural experiences and community gatherings that ultimately led to the *trampoline park party*.

Each edge in the narrative is verbalized in the opposite order to its place in the causal chain. All narratives can be found within the attached code and files.

As can be seen in Figure 2 (left), in the *Forward* direction a simple question prompting strategy asking causal questions (line labeled Forward Mean), naive CoT prompting (line labeled Forward-Chain Mean) where we ask the LLM to think step-by-step, and an In-Context Learning prompt (line labeled Forward-Context Mean) where we provide example narratives and causal answering all perform fairly well. This is in contrast to the *Reverse* orientation when we look at the performance of the simple prompt (line labeled Reverse Mean), naive CoT prompt (line labeled Reverse-Chain Mean), and In-Context Learning prompt (line labeled Reverse-Context Mean). From this plot, we can note that naive COT prompting and In-Context Learning prompts do not seem to boost accuracy under our conditions. Perhaps more interestingly, we find that the way the LLM answers questions using the narratives is not consistent with the causal graph $G'$ that the LLM builds when asked to predict the underlying graph structure (see consistency plot in bottom of Figure 2, where consis-

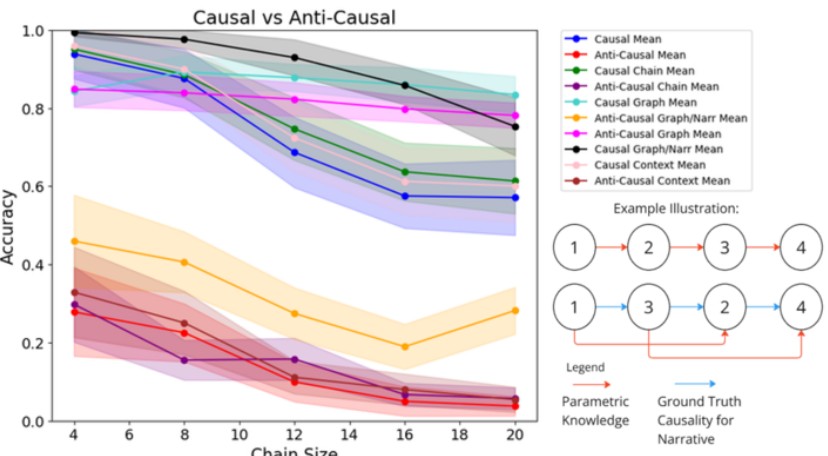

Figure 3: GPT-4 Test of the LLM's ability to reason on narratives that agree with parametric knowledge (Causal) and disagree with parametric knowledge (Anti-Causal). Label descriptions match those of Figure 2. Example illustration of how the $G$, the ground truth causality, is set up. Shaded zone around lines is 95 % Confidence Interval

tency measures agreement between the answers of the LLM and $G'$). Additionally, the trend of those prompting strategies on the consistency plot for the *Forward* orientation narratives (comparing performance to $G'$) mirrored their trend on the accuracy plot which compares performance to ground truth $G$ (left side).

This lead us to try using only $G'$ to answer causal questions and we found that it did significantly better in the *Forward* direction than the other prompting strategies (line labeled Forward Graph Mean), while doing no better in the *Reverse* direction (line labeled Reverse Graph Mean). In this case, once $G'$ is extracted by the LLM it is not given to the LLM again to answer questions (but rather used directly).

When prompted to use the narrative and $G'$ (the LLM is given $G'$ in this case in the prompt) to answer questions for *Forward* direction narratives, the accuracy again increases. This technique could be thought of as a type of CoT prompting strategy. In the Claude 3.5 Sonnet 4, The LLM also does far better with the *Forward* topological ordering than it does for the *Reverse* ordering. For the *Forward* ordering, the best performance again comes from using the extracted graph in concert with the narrative in the prompt.

### 3.3 PARAMETRIC KNOWLEDGE CONFLICT

We also find that LLMs tend to rely on parametric knowledge when it is present, and can fail when narratives run counter to their parametric knowledge. To test this, instead of picking random events, we have to draw out the LLM's pre-existing parametric knowledge. To do so, we prompt the LLM to pick a series of events such that each event has some relation to the subsequent event–either the event is *Causal* to the next event (e.g., disease causes shorter lives) or the event is *Anti-Causal* (e.g., disease causes longer lives). Let's call this graph of parametric knowledge $P$. We then take the odd indexed events (1st, 3rd etc) from $P$ and place them in the first half of the causal ground truth graph $G$ and the even indexed events (2nd, 4th etc) from $P$ in the second half of $G$. As we can see in the example illustration in Figure 3, each node in the first half of the causal graph $G$ now has a parametric relation to a specific node in the second half of $G$. For example we might know that node 1 is *Anti-Causal* to node 2 from parametric knowledge. Thus when we make the causal ground truth graph $(1 \rightarrow 3 \rightarrow 2)$, create a narrative from it, and then ask the LLM if node 1 causes 2 based on the narrative, it should say yes even though that disagrees with its parametric knowledge. After the ground truth graph is created, we generate the narrative in the *Forward* topological orientation to avoid confounding failure modes.

We find that, in synthetic experiments, this kind of occurrence is generally rare and the LLM finds the correct causal relation generally only when that relation agrees with its parametric knowledge.

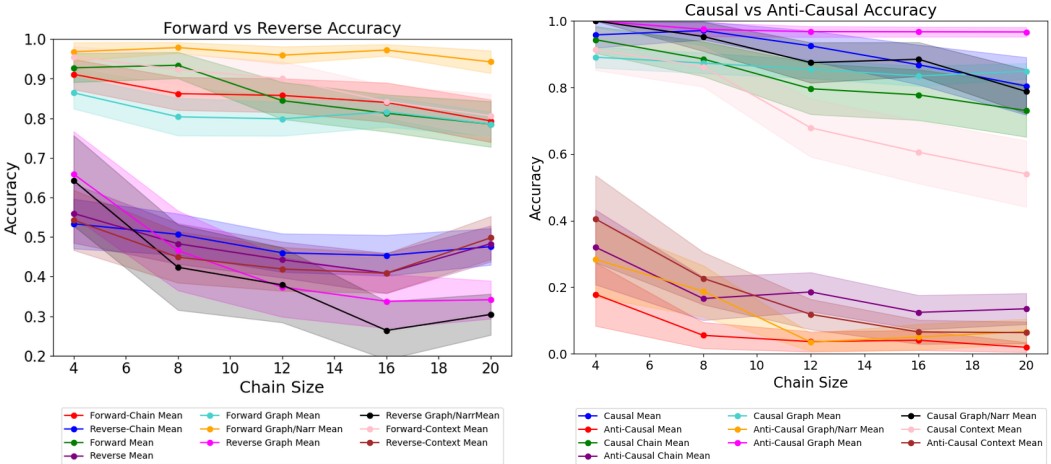

Figure 4: Claude 3.5 Sonnet Experiments: Left side plot mirrors the left side plot of Fig. 2 and the right side mirrors the plot in Fig. 3. Descriptions of prompt strategies match the before-mentioned figures.

This is exemplified in the plot in Figure 3 where we observe that when certain pairs of events are *Causal* in the parametric knowledge, the LLM also generally finds that the pairs of events in question are *Causal* given the narrative. However, when the pairs of events are *Anti-Causal* (like nodes 1 and 2 in our previous example), even though the LLM should find them to be *Causal* given the narrative it generally does not. For the *Causal* parametric knowledge case, prompting with just the extracted graph $G'$ is usually slightly better at most chain sizes than solely using the narrative (in simple prompts, COT or In-Context Learning), and this is only improved in most instances when using the narrative and extracted graph in the prompt.

We also notice an interesting phenomenon for the *Anti-Causal* parametric case where although using the extracted graph and narrative is better than any prompting strategy with just the narrative, using just the extracted graph provides massive improvements over all of these modalities and is comparable in performance to when the parametric knowledge is *Causal*. As an even more stark finding, the extracted graph for the Claude 3.5 Sonnet experiment in the right side of 4 has near perfect performance for the Anti-Causal case. It seems the narrative may only serve to distract the LLM when parametric knowledge disagrees with the narrative. When the narrative is provided, the LLM will often say things (in its COT explanation) like "Although this relationship is displayed in the chain of events in the narrative, it is not logical and counter-intuitive" and answer incorrectly. This happens despite explicit instruction to ignore outside information, allow for ill-logical relationships, and answer solely based on the hypothetical narrative.

## 3.4 FAILURE OF LONG-TERM CAUSAL REASONING

The performance of LLMs tends to decay as the size of the narrative and the number of events in the narrative increases. As we can see in Figures 2 and 3, generally all prompts that solely rely on the narrative without the extracted graph to answer causal questions have a performance that decays with the number of events in the narrative (chain size). The exception to this occurs in the *Reverse* topological orientation in 2 as the accuracy is psuedo-random at all chain sizes. This finding is supported by the well known phenomenon that LLMs often fail to reason about longer form contexts especially as they "get lost in the middle." (Liu et al., 2023). What perhaps is a surprising result that is the extracted graph tends to maintain a consistent level of accuracy for causal answering across narrative sizes even when only using the narrative in the prompt leads to severe dips in accuracy.

## 4 Experiments with Real World Causal Graphs

In this section, we test our methods on narratives constructed using real-world causal graphs from *CauseNet* (Heindorf et al., 2020), a large-scale knowledge graph of (claimed) causal relationships between real-world concepts. We use *GPT-4o* (OpenAI et al., 2024) for our experiments.

### 4.1 Generating the Narratives

The *CauseNet* dataset can be represented as a collection of $D$ tuples $\{(C_i, E_i, \mathbf{S}_i\}_{i=1}^D$, where $C_i$ denotes the cause (e.g., smoking), $E_i$ denotes the effect (e.g., disability), and $\mathbf{S}_i$ is a set of sentences (extracted from Wikipedia and ClueWeb12 (Callan, 2012)) that entail a causal relationship from $C_i$ to $E_i$. We retrieve causal chain graphs $V_1 \to V_2 \to \ldots \to V_N$ of various lengths, where each causal relation $V_i \to V_{i+1}$ is from *CauseNet*. Below, we describe the strategy for generating narratives for a given chain graph (see Appendix B.1 for the prompt templates).

**Semi-synthetic narratives.** Each chain graph is verbalized into a narrative by prompting the LLM to generate a sentence one edge at time such that the sentence for $V_i \to V_{i+1}$ logically follows the sentence for the previously verbalized edge. This ensures that the narrative as a whole is sensible. We generate the narratives by enumerating the edges in the topological order of the graph and in the reverse order. We call this semi-synthetic because the nodes in the graph represent real-world causal relationships but the narratives are synthetic. For example, the following is the generated narrative in the forward direction for the causal chain *fatigue → accidents → injury*:

> *Fatigue* can cloud judgment and slow reaction times, leading to an increase in *accidents* on the road. As a result, these *accidents* often lead to serious *injury* for those involved, highlighting the dangerous consequences of driving while fatigued.

**Real-world narratives.** For the real-world narratives, the sentence for each edge is chosen from the *CauseNet* dataset. To ensure that constructed narratives remain coherent, we prompt the LLM to ensure that the sentences for every pair of adjacent edges logically follow each other. For example, the following is the generated narrative for the causal chain *fatigue → accidents → injury*:

> Workers work long hours in mines and factories where *fatigue* and a lack of concentration can easily cause *accidents*. These *accidents* are the leading cause of *injury* in this country for people ages 1-34.

Additional examples of semi-synthetic and real-world narratives are presented in Appendix B.2 (the entire set of narratives used for our experiments is attached in the supplement).

### 4.2 Effect of Narrative Topological Ordering and Chain Length

As described in the previous section, we verbalize each causal chain graph $V_1 \to V_2 \to \ldots \to V_N$ from *CauseNet* into a narrative in the forward and reverse topological order. We give the narrative as input to the LLM and evaluate its causal reasoning abilities using the following prompting strategies (see Appendix B.3 for the prompt templates): (1) (Direct prompting) For every pair of nodes $(V_i, V_j)$, we ask does $V_i$ have a direct or indirect causal effect on $V_j$; (2) (CoT prompting) Same as (1), but we now use CoT and ask the LLM to think step-by-step; and (3) (Graph Extraction) Generate a causal chain graph for the narrative given the node identities $(V_1, V_2, \ldots, V_N)$ in a random order.

We evaluate the accuracy for each pair of nodes $(V_i, V_j)$ for the three prompting strategies on the semi-synthetic and real-world narratives (see Fig. 5). We denote the Direct prompting strategy (1) using the labels *Forward* and *Reverse*; the CoT prompting strategy (2) using the labels *Forward_COT* and *Reverse_COT*; and the accuracy using the extracted graph from strategy (3) using the labels *Forward_Graph* and *Reverse_Graph*.

In both the semi-synthetic (Fig. 5a) and real-world narratives (Fig. 5b), the *Forward_Graph* strategy performs the best, with its accuracy remaining stable even as the chain length increases. By contrast, *Reverse_Graph* has relatively lower accuracy, with the accuracy declining as the chain length

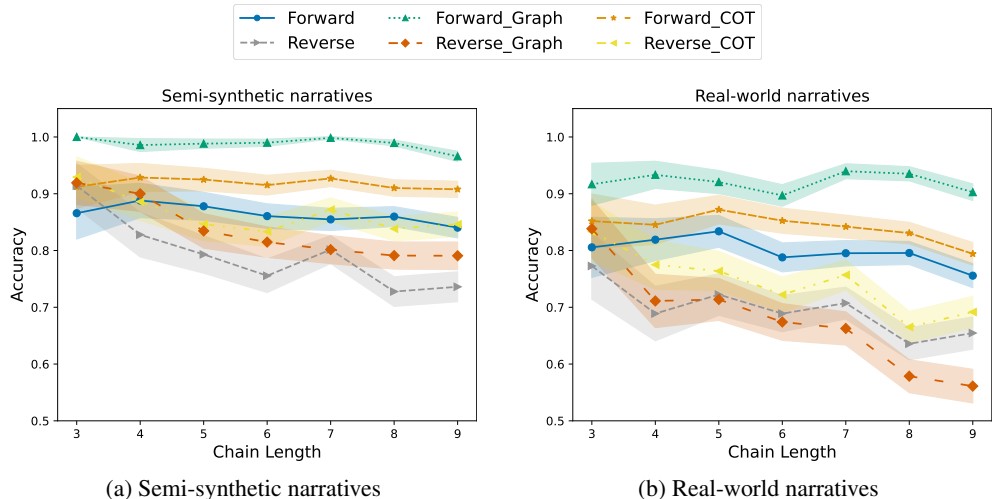

(a) Semi-synthetic narratives       (b) Real-world narratives

Figure 5: The accuracy of various prompting strategies (error bars denote 95% CIs). We observe that the accuracy is lower in the reverse direction (and tends to decay as the chains get longer).

|  | Forward (w CoT) | Reverse (w CoT) | Forward Graph | Reverse Graph |
|---|---|---|---|---|
| **Semi-synthetic** | | | | |
| Without Conflict | 97.86 (97.94) | 95.85 (96.47) | 98.54 | 87.99 |
| With Conflict | 65.47 (80.54) | 45.63 (67.83) | 97.70 | 84.32 |
| **Real-world** | | | | |
| Without Conflict | 95.54 (96.06) | 92.38 (91.63) | 87.74 | 57.93 |
| With Conflict | 48.03 (59.57) | 27.40 (49.43) | 85.67 | 53.09 |

Table 1: The average accuracy across different narratives with the three prompting strategies partitioned by whether the cause-effect pairs conflict with the LLM's parametric knowledge (we omit the 95% CIs as they are smaller than 0.3). The accuracy (with and without CoT) is substantially lower when knowledge conflicts exist, suggesting that LLM relies on this knowledge rather than the narrative. By contrast, the graph extraction is more robust to knowledge conflicts.

increases (this effect is particularly significant in the real-world narratives). We also observe that *Forward*(_COT) outperforms *Reverse*(_COT), with the *Reverse*(_COT) accuracy declining substantially as the chain size gets large. Additionally, unlike the forward direction, the graph extraction does not improve accuracy substantially over the Direct and CoT-based prompting strategies. Altogether, these results demonstrate that LLM's ability to extract the causal graph as well as determine causality between two nodes declines when the narrative is verbalized in the reverse order, with the impact of the increasing chain lengths being particularly pronounced in the reverse direction.

### 4.3 Effect of Parametric Knowledge Conflicts.

Next, we analyze the extent to which the LLM relies on its parametric knowledge to answer the causal query as opposed to the specific causal effects expressed in the narrative. For every pair of nodes $(V_i, V_j)$ in the chain graphs, we extract the parametric knowledge of the LLM by prompting the LLM to answer "Does $V_i$ typically have a causal effect on $V_j$?". This parametric knowledge represents some average case notion of causality that is learned by the LLM from the pretraining corpora.

Successful causal reasoning often requires going beyond the parametric causal knowledge as it can differ from the causality between events in the context of the narrative at hand. For example, in a chain graph from the our dataset, there is a causal path from *streambank erosion* to *higher prices*, but this contradicts the LLM's parametric knowledge since this causal effect may not typically exist

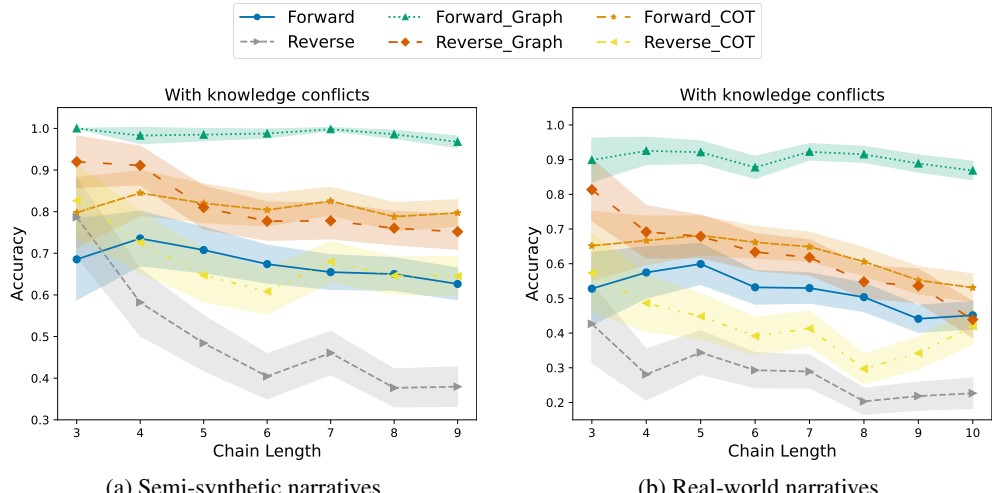

(a) Semi-synthetic narratives         (b) Real-world narratives

Figure 6: Results on the subset of cause-and-effect pairs that conflict with the LLM's parametric causal knowledge. We observe that while the graph extraction is robust to conflicts, the Direct and CoT prompting strategies suffer significantly (particularly in the reverse direction), indicating that the LLM returns its parametric knowledge instead of paying attention to the narrative.

in the real-world. In the chain graphs we evaluate, roughly 39% of cause-and-effect pairs conflict with the LLM's parametric knowledge.

We evaluate the three prompting strategies separately on the subsets of cause-and-effect pairs that are in agreement and in conflict with the parametric knowledge (see Table 1). We observe that when there is no conflict (i.e., the parametric knowledge agrees with the causality expressed in the narrative), the accuracies in both the *Forward* and *Reverse* directions (with and without CoT) are greater than 90%. However, when the parametric knowledge conflicts with the narrative's causality, the accuracy of *Forward* and *Reverse* (even with CoT) is significantly lower: even with CoT, the LLM has below-random accuracy in the reverse direction. This suggests that when asked to reason about cause and effect in a narrative, the LLM seems to rely heavily on its parametric knowledge and is unable to grasp the specific causal chains expressed in the narrative itself (despite the causal chains as a whole being realistic).

Interestingly, when using extracted graph for performing causal reasoning (*Forward Graph* and *Reverse Graph*), the performance in both cases seems to be comparable. This is likely because when asked to extract the graph from the narrative with the *given* set of nodes, the LLM pays more attention to the entire narrative as opposed to when directly queried on a cause-and-effect pair (where the LLM defaults to its parametric knowledge). These results show that even when the LLM constructs a reasonably good causal chain graph in the *Forward* direction, the LLM does not leverage this graph when queried directly about the causal effects in the narrative (even with CoT), further highlighting the advantage of extracting the causal graph directly.

Finally, we also observe that the accuracy of *Forward* and *Reverse* prompting strategies in Fig. 5 were largely driven by the parametric knowledge. To disentangle this effect, we evaluate the three prompting strategies across different chain lengths *only* on the subset of cause-and-effect pairs where a knowledge conflict exists (see Fig. 6). We observe that the accuracy of all prompting strategies, except *Forward_Graph* and *Reverse_Graph*, decline substantially relative to Fig. 5, with *Reverse* falling well below the random baseline of 50% as the chain length gets larger. In the real-world narratives (Fig. 6b), even with CoT, the accuracy does not improve over the random baseline of 50% in the reverse direction, demonstrating that the LLM ignores the causal chain in the narrative. Qualitatively, we observe that in the case of knowledge conflicts, the CoT explanation justifies the incorrect (according to the narrative) causal relationship by injecting its parametric knowledge into the chain-of-thought. Additionally, we also evaluate the three prompting strategies on the subset of cause-and-effect pairs in agreement with parametric knowledge (see Appendix B.4), showing high accuracy of the Direct and CoT-based prompting strategies on this subset.

## 5 DISCUSSION

Our work takes some initial strides towards examining the reasons behind the success and failure of LLMs to reason causally on narratives that express a chain of causal events. We focus on two questions of key importance in causality: (1) Does one event cause another? and (2) Can the LLM extract the causal graph from the narrative.

We find three significant failure modes of LLM reasoning by conducting experiments in carefully controlled synthetic, semi-synthetic and real-world settings:

1. Topological Ordering: LLMs tend to perform well when the ordering of events in the narratives matches that of the ordering of the underlying graph. For example if B is mentioned first in the narrative and then A, then the LLM would would expect $B \to A$ even if $A \to B$ is the truly underlying causal graph. As such, performance breaks down when the ordering of events in the narratives does not match that of the underlying graph.

2. Parametric Knowledge: LLMs rely on their parametric pre-training knowledge as a shortcut to infer causal relations. When the narrative suggests one causal relation (e.g., $A \to B$) and the parametric knowledge suggests another ($B \to A$), the LLM will often answer using its parametric knowledge, ignoring the narrative.

3. Narrative Size: LLM accuracy degrades as the chain length increasing, resulting in poor reasoning on longer narratives.

We also find that asking the LLM to extract what it predicts to be the underlying graph from the narrative, and using just that extracted graph or the extracted graph in concert with the narrative to answer can lead to mitigation of these failure modes.

Our work builds upon past work that examines causal reasoning in LLMs. Various works including Jin et al. (2023) suggest that LLMs may have difficulty performing causal reasoning in tasks involving numerical probabilistic computations. Previous work by Zečević et al. (2023) conjectured that LLMs simply imitate causal relations from their training data. Our work aims to assess the more colloquial kind of causal reasoning from textual narratives (not just on individual sentences) that is required to understand why certain events occurred and how they would impacted by certain interventions. Like previous works, we find a strong reliance on pre-training knowledge. However, we also suggest a way to mitigate this by using the LLM to extract the entire causal graph from the narrative at once. When doing this, the graph remains generally accurate even in the presence of conflicts with parametric pre-training knowledge.

### 5.1 LIMITATIONS AND FUTURE WORKS

One limitation of our work is that there are other forms of causal reasoning that we did not test for in the narratives. This motivates many potential directions for future work. For example, it could be interesting to ask the LLM to reason about counterfactual cases. Another limitation of our work is that we only deal with chain graphs. An interesting direction for future work would be to generate narratives from complex graph structures. This could potentially yield novel failure modes and insights into how LLMs reason. Another limitation of our work is the amount of insight we provide behind the phenomenon we observe. When we notice the graphs doing better than the prompts that only use the narrative itself, one conjecture is that this is because the LLM is looking at the entire narrative and reasoning about it to make the extracted graph–however we don't formally prove this. Future work could theoretically or empirically examine explanations for such LLM behavior and others noted in the paper.

Our analysis also has implications for algorithmic interventions to improve causal reasoning. The failure modes we identify in this paper could inform the design of targeted synthetic tasks to use in finetuning for improved causal reasoning. Additionally, our findings on the benefits of extracting a causal graph can inform prompt engineering efforts to elicit reliable causal reasoning from language models. We believe investigating both directions represents an exciting direction for future works.

## 6 REPRODUCIBILITY STATEMENT

Our experimental methodology primarily relies on prompting API-based large language models (in particular OpenAI GPT-4 and Anthropic Claude). We provide select sample prompts for each experiment in the corresponding Appendix section. Moreover, we will release a complete repository of our data (including all generated narratives and extracted graphs), processing scripts, and LLM prompting pipeline in the supplementary materials.

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

# Appendix

## A  SYNTHETIC DATA EXPERIMENTS

### A.1  SELECTED SYNTHETIC PROMPTS

Full list of prompts, data processing and results in attached code
$E$ is the list of events generated by the LLM
$G$ is the ground truth causal graph used to generate the narrative
$G'$ is the extracted graph from the narrative by the LLM
$N$ is the narrative generated by the LLM
$n$ is the number of events to generate
$A$ is the first item in the query and $B$ is the second so the question would be does $A$ cause $B$

#### A.1.1  TOPOLOGICAL EXPERIMENT - GENERATING RANDOM EVENTS ($E$)

"generate $n$ random distinct events"

#### A.1.2  PARAMETRIC EXPERIMENT -GENERATING CAUSAL EVENTS ($E$)

"generate $n$ events that cause each other (for example the first event causes the second and the second causes the third)"

#### A.1.3  PARAMETRIC EXPERIMENT - GENERATING ANTI-CAUSAL EVENTS ($E$)

"generate $n$ events that are anticausal, for example the first event could be cancer and the second event could be a longer life because in reality, cancer causes a shorter life. Make it so each of the $n$ events are anticausal in this way such that the next event is actually the opposite of what it should be. For 3 events we might have 1.Rain 2.Plants Die 3.Increased Oxygen"

#### A.1.4  FORWARD TOPOLOGICAL NARRATIVE ($N$)

" context is $G$, generate a hypothetical narrative from this causal chain graph and make causal relations explicit, even when the causal relations do not make sense, keep the causal relations as they were in the context. The events in the story should occur in the same order as in the chain graph (eg first item in chain graph should appear in narrative before the second item)"

#### A.1.5  REVERSE TOPOLOGICAL NARRATIVE ($N$)

"context is $G$, generate a hypothetical narrative from this causal chain graph and make causal relations explicit, even when the causal relations do not make sense, keep the causal relations as they were in the context. The events in the story should occur in the opposite order as in the chain graph (eg last item in chain graph should appear first in the narrative etc)"

#### A.1.6  STANDARD PROMPT

"Use this narrative $N$ as context. Did $A$ cause $B$? Output your answer with $<answer> Yes/No </answer>$. The cause can be direct or indirect."

#### A.1.7  CHAIN OF THOUGHT PROMPT

"Use this narrative $N$ as context. Did $A$ cause $B$? Do step by step reasoning. Then output your answer with $<answer> Yes/No </answer>$. The cause can be direct or indirect."

#### A.1.8  IN-CONTEXT PROMPT

"Use this narrative $N$ as context. Did $A$ cause $B$? Output your answer with $<answer> Yes/No </answer>$. The cause can be direct or indirect. An example narrative would be: Rains leads to plants growing. This then causes increased oxygen in the atmosphere. A potential question

would be does rain cause increased oxygen in the atmosphere? The answer would be Yes. Another example narrative would be: Increased oxygen in the atmosphere is because of plants growing. Plants grow because rain provides them essential nutrients. A potential question would be does rain cause increased oxygen in the atmosphere? The answer would be Yes. Another example narrative would be: Rain leads plants to grow. Plants growing causes less oxygen in the atmosphere. A potential question would be does rain cause less oxygen in the atmosphere? The answer would be Yes."

### A.1.9 NARRATIVE + GRAPH PROMPT

"Use this narrative $N$ and this causal ordering $G'$ ((such that each item is a cause of every item after it, for example the first list item is a cause of the third, fourth, fifth items etc)) as context. Did $A$ cause $B$? Output your answer with $< answer > Yes/No < /answer >$. The cause can be direct or indirect."

## B REAL-WORLD CAUSAL GRAPHS

### B.1 PROMPT TEMPLATES FOR NARRATIVE GENERATION

Recall that we have a ground truth causal chain graph of the form $V_1 \rightarrow V_2 \rightarrow \ldots \rightarrow V_N$ from *CauseNet* that we need to verbalize into a coherent narrative. For the semi-synthetic narratives, we use the LLM (GPT-4o) to do so one edge at a time, while ensuring that the newly verbalized edge logically follows the previous one. The following is the prompt template for generating the narratives in the topological order of the graph:

> Output a short narrative (use one or two sentences) that expresses the causal link
> $[V_i \rightarrow V_{i+1}]$ and logically follows this narrative:
> { Narrative for the previous edge $V_{i-1} \rightarrow V_i$}.
> Ensure that the combined sentences convey the causal chain $[V_{i-1} \rightarrow V_i \rightarrow V_{i+1}]$
> and that the words $[V_i, V_{i+1}]$ are present. Only output the newly generated narrative.

Similarly, we generate narratives in the reverse topological order of the graph by verbalizing edges in the reverse direction with the following prompt template:

> Output a short narrative (use one or two sentences) that expresses the causal link
> $[V_i \rightarrow V_{i+1}]$ and logically follows this narrative:
> { Narrative for the previous edge $V_{i+1} \rightarrow V_{i+2}$}.
> Ensure that the combined sentences convey the causal chain $[V_i \rightarrow V_{i+1} \rightarrow V_{i+2}]$
> and that the words $[V_i, V_{i+1}]$ are present. Only output the newly generated narrative.

For generating real-world narratives, for each edge $V_i \rightarrow V_j$, we use the set of sentences from *CauseNet*. Note that each edge in *CauseNet* is linked to multiple sentences from various sources. Picking a sentence for each edge at random and concatenating them does not always lead to sensible narratives. To improve the quality of narratives, we use the following prompt to concatenate sentences for adjacent edges:

> Consider the following sentences.
> { Sentence for edge $V_i \rightarrow V_{i+1}$ }. { Sentence for edge $V_{i+1} \rightarrow V_{i+2}$ }.
> Do the sentences logically follow each other and express the causal chain $[V_i \rightarrow V_{i+1} \rightarrow V_{i+2}]$? Answer with Yes or No.

For verbalizing narratives in the topological order, for a given graph $V_1 \rightarrow V_2 \rightarrow \ldots \rightarrow V_N$, we only use sentences such that the above prompt returns *Yes* for every pair of adjacent edges $V_i \rightarrow V_{i+1} \rightarrow V_{i+2}$. This ensures that the narrative as a whole remains coherent and conveys the entire causal chain graph. We use a similar prompting strategy to verbalize narratives in the reverse topological order.

## B.2 ADDITIONAL EXAMPLES OF THE GENERATED NARRATIVES

### B.2.1 SEMI-SYNTHETIC NARRATIVES

Below, we present some examples of semi-synthetic narratives in the forward and reverse directions.

The narrative in the forward direction for the chain *higher prices → reduced demand → lower prices*:

> As *higher prices* swept through the market, consumers began to tighten their budgets, leading to a noticeable *reduction in demand* for many goods. As a result of the *reduced demand*, suppliers were forced to *lower prices* in order to attract buyers back to the market.

The narrative in the reverse order for the causal chain *bankruptcy → bad credit → rejection → anger*:

> The sting of rejection ignited a fire within her, transforming her hurt into a seething anger that demanded to be felt. Her bad credit had led to the rejection she never saw coming, and now that sting of rejection ignited a fire within her, transforming her hurt into a seething anger that demanded to be felt. Her bankruptcy had left her with bad credit, a shadow that loomed over her every application, and now that sting of rejection ignited a fire within her, transforming her hurt into a seething anger that demanded to be felt.

The narrative in the reverse order for the causal chain *pollution → climate change → extreme weather events → natural disasters*:

> As extreme weather events become more frequent and severe, they increasingly lead to devastating natural disasters that disrupt communities and ecosystems alike. Climate change is driving the rise in extreme weather events, which in turn are causing unprecedented natural disasters that threaten the stability of communities and the health of ecosystems. Pollution is a major contributor to climate change, which is driving the rise in extreme weather events that threaten the stability of communities and the health of ecosystems.

### B.2.2 REAL-WORLD NARRATIVES

Below, we present some examples of real-world narratives in the forward and reverse directions.

The narrative in the forward direction for the chain *higher prices → reduced demand → lower prices*:

> *Higher prices* generally lead to reduced demand. *Lower prices*, caused by *reduced demand* and increased competition for soybeans and corn, largely contributed to the overall bulk export decline.

The narrative in the reverse order for the causal chain *bankruptcy → bad credit → rejection → anger*:

> Embittered by an abusive upbringing, seething with resentment, irritated by others' failure to fulfill his or her superior sense of entitlement, and fuelled by anger resulting from rejection, the serial bully displays an obsessive, compulsive and self-gratifying urge to displace their uncontrolled aggression onto others whilst exhibiting an apparent lack of insight into their behavior and its effect on people around them. Bad credit normally leads to rejection but now with bad credit secured loan, you can avail the loan of your choice. For example, if you are applying for a loan, the lender may reject your application on the basis of bad credit caused by bankruptcy.

The narrative in the reverse order for the causal chain *pollution → climate change → extreme weather events → natural disasters*:

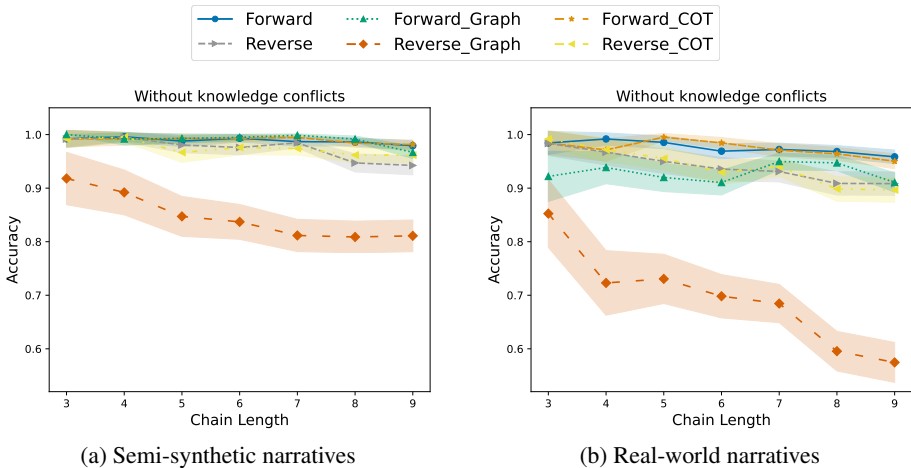

(a) Semi-synthetic narratives       (b) Real-world narratives

Figure 7: Results on the subset of cause-and-effect pairs that are in agreement with the LLM's parametric causal knowledge. We observe that the Direct and CoT prompting strategies have high accuracy for both semi-synthetic and real-world narratives.

> In addition to forced migrations from rising seas, climate change is also increasing extreme weather events causing natural disasters such as cyclonic storms (hurricanes or typhoons), floods and droughts. This is worsened by extreme weather events caused by climate change. This landmark bill would jump start the economy by creating millions of new clean energy jobs, increase national security by reducing dependence on foreign oil, and preserve the planet by reducing the pollution that causes climate change.

### B.3 PROMPT TEMPLATES FOR ASSESSING CAUSAL REASONING

We use the following template for the Direct prompting strategy:

> Consider the following hypothetical narrative.
>
> {narrative}
>
> According to the hypothetical narrative, does {cause} have a (direct or indirect) causal effect on {effect}? Answer in Yes/No.

We use the following template for the Chain-of-Though (CoT) prompting strategy:

> Consider the following hypothetical narrative.
>
> {narrative}
>
> According to the hypothetical narrative, does {cause} have a (direct or indirect) causal effect on {effect}? Think step-by-step and end your answer with <answer>Yes/No</answer>.

We use the following template to extract a chain graph from the narrative:

> Consider the following hypothetical narrative.
>
> {narrative}
>
> According to the hypothetical narrative, construct a causal chain graph using the following nodes: { nodes in random order }. Ensure that the graph contains all the given nodes and only output a single chain graph of the form <graph>node1 → node2 → node3 </graph>. Only output the graph between the <graph></graph>tags.

## B.4    ADDITIONAL RESULTS

In Fig. 7, we present the accuracy of the three prompting strategies discussed in Sec. 4 on the subset of cause-and-effect pairs that are in agreement to the LLM's parametric knowledge (this is a counterpart to Fig. 6). We observe that when the parametric knowledge agrees, the Direct and CoT-based prompting leads to high accuracy.

