# OpenReview forum: "Failure Modes of LLMs for Causal Reasoning on Narratives"
_ICLR.cc/2025/Conference — ICLR 2025 Conference Withdrawn Submission_

### Official Review · Reviewer_Ah2k · 2024-11-01

**Soundness:** 2
**Presentation:** 2
**Contribution:** 2
**Rating:** 3
**Confidence:** 3

**Summary:**

In their work, the author aim to inspect the causal reasoning capabilities of LLMs in natural language texts. The authors study effects on predictive performance in long-term reasoning and investigate performance degradation when reason over contextual information that is counterfactual to their learned 'parametric' knowledge.

The authors utilize LLMs to generate (semi-synthetic) narratives about events resembling causal chains from CauseNet. In that, 'anti-causal' relations are purposefully injected, which stand in contrast to factual observations of the real-world. Finally, reverse topological narratives are created in which items of the causal chain appear in reverse order.

The authors measure LLM performance, by tasking the LLMs to restore the causal graph G from the narrative, using CoT and In-Context learning. In summary, the authors find that performance degradates when presenting narratives in reverse causal order, for 'anti-causal' relations and with longer chains of events. CoT and in-context learning do not boost performance, while providing graph information does seem to help even for longer chains.

**Strengths:**

The authors provide a series of evaluations on generated, semi-synthetic and real-world causal relations, that sufficiently support the claims of the paper. The paper provides a more detailed analysis on causal reasoning capabilities than previous works and experiments are generally well described.

By reasoning over counterfactual observations, the authors successfully show that LLM have trouble incorporating contextual information of the prompt that stands in contrast to learned knowledge. While previous work already identified general shortcomings in causal reasoning capabilities of LLM, the authors provide a more detailed breakdown of particular factors. The found confounding between predictive performance and order of narrative might be particular important to overcome.

While the authors test LLMs in a rather restricted settings, the presented results provide clear evidence towards the claimed effects. Limitations on the causal graph structure and types of inspected effects are discussed sufficiently, and one would expect results to only worsen when applied to more complex graphs.

**Weaknesses:**

While I agree on many interpretations of the presented results, I have several questions regarding the soundness of some experimental setups and the provision of graph information:

1) In section 3.2, the authors give an example of a "Film festival → Food truck rally → Trampoline park party" narrative. While the goal of generating a reversed narrative was pursued, the term 'Trampoline park party' was again mentioned after the 'Film festival' in the last sentence, establishing a relation in the correct relation order. Enforcing items of the causal chain to appear in reversed order, might pose a challenging task, as item positions can get switched due to the syntax of natural language. It would be interesting to know, how often such generation 'errors' occur within the data. The paper could be improved by analyzing to which extend LLM generations obey the desired ordering of events.
2) Throughout the paper, the authors refer to 'anti-causal' effects to indicate inversion of the causal effect *strength*, e.g. "cancer->longer life". The term 'anti-causal', is commonly reserved to indicate a reversed causal *direction* e.g. "cancer<-longer life". This is quite confusing, and given that it appears in the prompt presented in A.1.3, might have an impact on narrative generation. I would like to suggest to consider an alternative wording, e.g. counterfactual effects.
3) The authors provide causal graph information to the LLM during several experiments. While this seems to improve LLM performance, I could not find any details on the exact format of the provided graph information. It could be, that graph information is provided in the correct causal order, which would again improve LLM performance due to ordering bias and not due to reasoning capabilities of the LLM. The paper could be improved by further specifying and, if needed, randomizing the order of the provided causal effects in the experiments.
4) The reordering of relations in the experimental setup of section 3.3 might conceal the true influence of the anti-causal edges on the predictive performance of LLMs. As reordering was previously shown to reduce performance, it becomes unclear whether anti-causality or reordering of relation is the true cause for the performance degradation. I believe that effects of anti-causal relations might be better demonstrated when keeping the ordering intact and simply varying the amount of anti-causal relations within the chains.
5) To the best of my understanding, the problem of identifying and reasoning about causal relations in texts is commonly known as event causality identification (ECI; in the field of NL) or simply event causality (when moving closer to classical Pearlian causality). While the authors are able to produce new insights on the failure modes of LLM, several prior works derived similar results and are not discussed. This regards general reasoning with counterfactual contexts (e.g. [1,2,3]), as well as general causal text understanding (e.g. [4,5]). While these works certainly form a niche within the bigger field of causality and LLMs, I would recommend the authors to consider these relations.


**Minor**

* Unnecessary brackets are added for citations in Sec. 3.1 (l156/157) and in Sec. A.1.9.
* The resolution of figure 3 is quite low, such that the legend is unpleasant to read. I would like to suggest to either embed the figure as a vector graphic/PDF or increase its resolution.
* Some words seem to be missing in line 248.


[1] Li, Jiaxuan, Lang Yu, and Allyson Ettinger. "Counterfactual reasoning: Do language models need world knowledge for causal inference?." NeurIPS 2022 Workshop on Neuro Causal and Symbolic AI (nCSI). 2022.
[2] Li, Jiaxuan, Lang Yu, and Allyson Ettinger. "Counterfactual reasoning: Testing language models' understanding of hypothetical scenarios." arXiv preprint arXiv:2305.16572 (2023).
[3] Frohberg, Jörg, and Frank Binder. "Crass: A novel data set and benchmark to test counterfactual reasoning of large language models." arXiv preprint arXiv:2112.11941 (2021).
[4] Gao, Jinglong, et al. "Is chatgpt a good causal reasoner? a comprehensive evaluation." arXiv preprint arXiv:2305.07375 (2023).
[5] Ashwani, Swagata, et al. "Cause and Effect: Can Large Language Models Truly Understand Causality?." arXiv preprint arXiv:2402.18139 (2024).

**Questions:**

My questions mainly regard the points mentioned in the weaknesses above. In particular, I would like to ask the authors to comment on the following questions:

1. I would like to ask the authors to comment on point 1 of the above weaknesses. In particular, I would like to know whether the generated narratives adhere to the underlying causal structure and ordering. Are there differences in predictive performance of the LLM for samples that deviate from the desired ordering?
2. Regarding point 3 of the above weaknesses: Does the order of the provided graph information cohere to the ordering of the causal chain and would this be suited to leak information to the model, possibly improving its performance? Could the authors please elaborate and provide evidence for or against this scenario?
3. I would like to ask the authors elaborate on the necessity of reordering events in the experiments of section 3.3. How can the effects of reordering and anti-causal edges be better differentiated?

---

### Official Review · Reviewer_NqdV · 2024-11-02

**Soundness:** 2
**Presentation:** 1
**Contribution:** 1
**Rating:** 3
**Confidence:** 4

**Summary:**

This paper investigates the causal reasoning abilities of Large Language Models (LLMs) in narratives. It identifies several limitations, including reliance on event topological order, struggles with long context, and over-reliance on parametric knowledge. Furthermore, the authors demonstrate that explicitly prompting LLMs to generate causal graphs can improve performance.

**Strengths:**

1. The paper tackles the crucial issue of LLM's causal reasoning ability, a topic of significant current interest.
2. The experiment includes both synthetic and real-world data, which provides a more comprehensive assessment.

**Weaknesses:**

1. The core contributions and findings of this paper appear to heavily overlap with the previously published work *LLMs Are Prone to Fallacies in Causal Inference*, which **appeared on arXiv in June 2024** and was subsequently **published at EMNLP 2024**. However, this highly relevant and similar prior work is not cited or discussed. Moreover, the technical details and experimental validation are less comprehensive than in the aforementioned paper. Ensuring proper citation of relevant prior work is crucial for advancing academic discourse and providing a transparent foundation for new contributions.
2. The evaluation uses yes/no questions, meaning the random guess baseline is 0.5.  It is concerning that many results for GPT-4, GPT-4o, and Claude-3.5-Sonnet are below this baseline. While this might be theoretically possible under certain circumstances, the paper does not provide sufficient explanation or analysis of this phenomenon.  This lack of explanation weakens the validity of the experimental findings and necessitates further investigation and a detailed error analysis.
3. The paper lacks essential details about the evaluation dataset, making it difficult to assess the validity of the results.  Crucially, the number of samples tested, the number of nodes in each graph, and the specific types of events present in the narratives are not clearly specified. While line 149 mentions that "events are real-world phenomena", this description is too broad and lacks necessary citations.  Furthermore, the inconsistency between Figure 5(b) (maximum chain length 9) and Figure 6(b) (maximum chain length 10) raises questions about whether different datasets were employed for the synthetic and real-world narratives, or is this an oversight? Clarification on these points is essential.
4. The paper does not adequately address how the quality of the generated narratives is ensured. How are the narratives verified to ensure they conform to the specified chain structures? What mechanisms are in place to check and assess the data quality? Furthermore, given the identified issue of parametric knowledge conflicts within the models, how does the generation process prevent the introduction of factual errors or inconsistencies into the narratives? This lack of detail raises concerns about the reliability of the experimental data.
5. While the paper's focus on chain causal relationships provides a starting point, the scope remains relatively narrow. Exploring more complex scenarios is crucial for a more comprehensive understanding of LLM's causal reasoing ability.
6. The model selection is quite limited, using only GPT-4, GPT-4o, and Claude-3.5-Sonnet for both data generation and evaluation. This raises concerns about the generalizability of the findings. Including leading open-access models, such as Llama 3.1, would strengthen the analysis and provide insights into whether the observed phenomena are specific to limited-access models or hold across a wider range of LLMs.
7. The paper's logical structure could be improved to enhance clarity and readability. Additionally, the writing also requires significant revision to ensure the work is accessible and understandable to the reader.
8. The poor quality of the figures hinders understanding of the presented work. For example, Figure 3 is blurry and difficult to interpret.
9. The referencing of both articles and figures needs substantial improvement. Several errors were noted, including incorrect article citations on lines 156-157 and incorrect figure references on lines 248, 304, and 320.

**Questions:**

Please refer to the 'Weaknesses' section.

---

### Official Review · Reviewer_M8kG · 2024-11-03

**Soundness:** 2
**Presentation:** 3
**Contribution:** 2
**Rating:** 5
**Confidence:** 3

**Summary:**

In this paper, the authors study the effectiveness of using LLMs to learn the cause effect relationships in a chain graph in both synthetic and real-world settings. The authors report the various modes through which the LLMs fails in this task. These failure modes can be summarized as follows:
1. LLMs rely heavily on the order in which the causal relationships are verbalized in the narrative.
2. LLMs use their parametric knowledge as a shortcut to answer causal questions.
3. LLMs fail more often when the narrative becomes larger.
The authors also investigate various prompting strategies that help avoid some of these pitfalls.

**Strengths:**

- The paper provides a formal characterization of the failure modes of an LLM towards causal reasoning tasks.
- Some of these failure modes are quite interesting, for example, the one where the narrative introduces the causal variables in reverse order leads to poor performance.
- The paper is well presented and written, making it easy to follow.

**Weaknesses:**

- In all the experiments, the narratives provided to the LLM convey significant insight on the causal structure of the chain graph under consideration. Using phrases like “results in”, “leads to”, “causes” in the narrative directly places a causal link between the variables. This approach would fail in scenario where such information is not already available. This makes the role of LLM quite straightforward (at least in the forward narrative setting).
- There was no explicit characterization of the error in the chain graph G’ that is produced by the LLM, without this it makes it unclear as to where the error stem in this setting stems from.
- It would help to provide more insight in to why there are inconsistencies between the answers to the causal reasoning task and the chain graph G’ predicted by the LLM
-in light of existing works in use of LLM for causal discovery, some of these failures have been observed by past works and hence advocated hybrid approaches by combining with data driven methods.

**Questions:**

- In Figure 4, Claude showcases a more significant divide between the forward and reverse settings for the narratives compared to ChatGPT, any insight as to why that is the case?
- What would happen if the narrative doesn’t explicitly state the causal relations using phrases like “results-in”, “causes”?
- What was the accuracy of LLM recovering the correct chain graph in the settings “Forward/Reverse-Graph”, “Causal/Anti-Causal Graph”?
- Why not combine both chain of thought reasoning and G’ to see if there is any improvement in performance?
-clearly prompt based solutions fails across many reasoning tasks as reported heavily by past works, and causal discovery is no exception.   Several past works suggest to combine the data driven method with LLM for causal discovery. Relying entirely on inherent knowledge of LLM have proven to be problematic in reasoning tasks and not sure this paper puts significant lights on as how we devise better methods for causal reasoning,

---

### Official Review · Reviewer_PNEG · 2024-11-03

**Soundness:** 3
**Presentation:** 3
**Contribution:** 3
**Rating:** 6
**Confidence:** 4

**Summary:**

The paper investigates the causal reasoning capabilities of recent LLMs; through various settings; via synthetic, semi-synthetic and real world narratives, comparing their consistency in forward and reverse causal narrative, against their parametric knowledge (pre-trained model knowledge) via  direct prompting, CoT prompting,  and graph extracting, with varying length of causal (anti-causal) chains. Results suggest that LLMs show considerable  bias towards their parametric knowledge, and are more successful in queries that align with them. They do in general perform pretty weak in causal reasoning task. The  extracted graph mostly leverages their strength since it helps them to spend more attention to the whole of the narrative text.

**Strengths:**

- Very relevant line of research, with high impact. (spotting the failure modes of LLMs on causal reasoning ).
-Good writing and. exposition in general and easy to follow.
- The proposed setting of approach is interesting, and the depth and multitude of the analysis is sufficient.
-Overall, results are interesting, intuitive and worth considering. I find the results expected but not trivial.  I am particularly pleased to see the positive effect of the causal graph, which can help the CoT practice as well, as they put it in the limitations sections.

**Weaknesses:**

My main concern  is that the experiments/results can be misinterpreted and therefore may not be strongly conclusive for the intended tasks,  due to the followings:

1) Vagueness in expressions: This is mainly due to vagueness in expression formulation sespecially in the reverse part. : Take the given example given in the paper: "The film festival set the stage for the rest of the events, creating a desire for cultural experiences and community gatherings that ultimately led to the trampoline park party."  Here the "The film festival set the stage for the rest of the events" sounds as if it took place afterwards due to "rest" of the events, and the part (desire and community gatherings) of the sentence mentioned before the "trampoline park party" races against the "film festival" and distracts the attention, hence the overall degrading in the causal reasoning estimation is likely.

2) Conformity to human communication: Another aspect is that LLMs can mis-read your intention, since they can falsely "assume" or "expect" you do typos or miswrite/communicate. This is a strategic choice in their design (and perhaps even their nature due to their training corpora) to not make them brittle, and work with real human conversations.


3) Referral mismatch. You say: “Although this relationship is displayed in the chain of events in the narrative, it is not logical and counter-intuitive” and answer incorrectly." Here, although you told the LLMs to ignore the parametric knowledge (or as you put it "explicit instruction to ignore outside information, allow for ill-logical relationships, and answer solely based on the hypothetical narrative"), it may still refer to the "meta-explanation" when it says it is not logical or counter-intuitive since it "informs" the user.  So there might be a spurious mismatch between the task you expect vs.  the setting LLM assume itself in.

4) On conflicts (unknown vs. opposite): You say: "streambank erosion to higher prices, but this contradicts the LLM’s parametric knowledge since this causal effect may not typically exist"  Here again I find this rather vague conflict, and LLM might likely to conform the given example, when it cannot find counter examples, and can simply hallucinate. I would expect rather a hard conflict e.g.,  inverse causation, something like, rooster cause the sun rise.

I appreciate the paper's work, but the above concerns is due to inherent complexity of dealing with natural language, and its discrepancy against the intended task (hence we should be ourself causally-careful.)

Typos:

-  would would
-the our dataset

**Questions:**

1) I wonder your takes on the issues I point out above.

2) Moreover, you use  GPT-4o only in real world experiments, why is that?

3) Why does the overall  accuracy scores decrease in Figure 5, in real world narratives compared to Semi-synthetic narratives?

---

### Author Response · Authors · 2024-11-25
**Withdrawal**

Hi reviewers, we would like to thank you for your commentary on our paper. We are appreciative of the time taken to provide constructive criticism, and review the contributions and potential shortcomings of the paper. In order to take the time to properly update our paper and model, we will withdraw our submission and work on addressing the concerns that have been brought up as well as work on clarifying how our paper provides a different contribution from the papers you have mentioned. Thanks so much.

---

### Note · Authors · 2024-11-25

**Comment:**

Hi reviewers, we would like to thank you for your commentary on our paper. We are appreciative of the time taken to provide constructive criticism, and review the contributions and potential shortcomings of the paper. In order to take the time to properly update our paper and model, we will withdraw our submission and work on addressing the concerns that have been brought up as well as work on clarifying how our paper provides a different contribution from the papers you have mentioned. Thanks so much.

**Withdrawal Confirmation:**

I have read and agree with the venue's withdrawal policy on behalf of myself and my co-authors.